# Quick sequential organ failure assessment score combined with other sepsis-related risk factors to predict in-hospital mortality: Post-hoc analysis of prospective multicenter study data

Ryo Ueno[1,2], Takateru Masubuchi[1], Atsushi Shiraishi [3]*, Satoshi Gando[4,5], Toshikazu Abe[6], Shigeki Kushimoto [7], Toshihiko Mayumi[8], Seitaro Fujishima[9], Akiyoshi Hagiwara[10], Toru Hifumi[11], Akira Endo[12], Takayuki Komatsu[13], Joji Kotani[14], Kohji Okamoto[15], Junichi Sasaki [16], Yasukazu Shiino[17], Yutaka Umemura[18]

1 Department of Intensive Care Medicine, Kameda Medical Center, Kamogawa, Chiba, Japan, 2 Australian and New Zealand Intensive Care Research Center, School of Public Health and Preventive Medicine, Monash University, Melbourne, Australia, 3 Emergency and Trauma Center, Kameda Medical Center, Kamogawa, Japan, 4 Department of Acute and Critical Care Medicine, Sapporo Higashi Tokushukai Hospital, Sapporo, Japan, 5 Division of Acute and Critical Care Medicine, Hokkaido University Graduate School of Medicine, Sapporo, Japan, 6 Department of Health Services Research, Faculty of Medicine, University of Tsukuba, Tsukuba, Japan, 7 Division of Emergency and Critical Care Medicine, Tohoku University Graduate School of Medicine, Sendai, Japan, 8 Department of Emergency Medicine, School of Medicine, University of Occupational and Environmental Health, Kitakyushu, Japan, 9 Center for General Medicine Education, Keio University School of Medicine, Tokyo, Japan, 10 Department of Emergency Medicine, Niizashiki Chuo General Hospital, Niiza, Japan, 11 Department of Emergency and Critical Care Medicine, St. Luke's International Hospital, Tokyo, Japan, 12 Trauma and Acute Critical Care Medical Center, Tokyo Medical and Dental University Hospital of Medicine, Tokyo, Japan, 13 Juntendo University Nerima Hospital, Tokyo, Japan, 14 Division of Disaster and Emergency Medicine, Department of Surgery Related, Kobe University Graduate School of Medicine, Kobe, Japan, 15 Department of Surgery, Center for Gastroenterology and Liver Disease, Kitakyushu City Yahata Hospital, Kitakyushu, Japan, 16 Department of Emergency & Critical Care Medicine, Keio University School of Medicine, Tokyo, Japan, 17 Department of Acute Medicine, Kawasaki Medical School, Kurashiki, Japan, 18 Department of Traumatology and Acute Critical Medicine, Osaka University Graduate School of Medicine, Osaka, Japan

* siris.accm@tmd.ac.jp

## Abstract

This study aimed to assess the value of quick sequential organ failure assessment (qSOFA) combined with other risk factors in predicting in-hospital mortality in patients presenting to the emergency department with suspected infection. This post-hoc analysis of a prospective multicenter study dataset included 34 emergency departments across Japan (December 2017 to February 2018). We included adult patients (age ≥16 years) who presented to the emergency department with suspected infection. qSOFA was calculated and recorded by senior emergency physicians when they suspected an infection. Different types of sepsis-related risk factors (demographic, functional, and laboratory values) were chosen from prior studies. A logistic regression model was used to assess the predictive value of qSOFA for in-hospital mortality in models based on the following combination of predictors: 1) qSOFA-Only; 2) qSOFA+Age; 3) qSOFA+Clinical Frailty Scale (CFS); 4) qSOFA+Charlson Comorbidity Index (CCI); 5) qSOFA+lactate levels; 6) qSOFA+Age+CCI+CFS+lactate levels. We

**Data Availability Statement:** Data cannot be shared publicly because of confidentiality. Data are



available from the Japanese Association for Acute Medicine (JAAM) SPICE study group/Ethics Committee for researchers who meet the criteria for access to confidential data. Please contact the corresponding author and/or the ethics committee (contact via jaam-6@bz04.plala.or.jp).

**Funding:** This work is supported by the Masason Foundation (grant number: AM000010). The funding body played no role in the study design; data collection, management, analysis, or interpretation; manuscript preparation; or the decision to submit the report for publication.

**Competing interests:** The authors have declared that no competing interests exist.

calculated the area under the receiver operating characteristic curve (AUC) and other key clinical statistics at Youden's index, where the sum of sensitivity and specificity is maximized. Following prior literature, an AUC >0.9 was deemed to indicate high accuracy; 0.7–0.9, moderate accuracy; 0.5–0.7, low accuracy; and 0.5, a chance result. Of the 951 patients included in the analysis, 151 (15.9%) died during hospitalization. The AUC for predicting in-hospital mortality was 0.627 (95% confidence interval [CI]: 0.580−0.673) for the qSOFA-Only model. Addition of other variables only marginally improved the model's AUC; the model that included all potentially relevant variables yielded an AUC of only 0.730 (95% CI: 0.687–0.774). Other key statistic values were similar among all models, with sensitivity and specificity of 0.55−0.65 and 0.60−0.75, respectively. In this post-hoc data analysis from a prospective multicenter study based in Japan, combining qSOFA with other sepsis-related risk factors only marginally improved the model's predictive value.

## Introduction

Sepsis is an overwhelming host reaction to infection leading to life-threatening organ failure. It is associated with high rates of mortality and morbidity. In 2016, sepsis was defined based on changes to the sequential organ failure assessment (SOFA) score; concurrently, the quick SOFA (qSOFA) score was introduced as a simple bedside tool to consider the possibility of sepsis in non-intensive care unit (ICU) settings [1].

The landmark study conducted by Seymour and his colleagues [1] compared the prognostic value of qSOFA with that of SIRS (Systemic Inflammatory Response Syndrome) score; this was followed by various validation studies [2–4] that assessed the predictive value of qSOFA for in-hospital mortality in patients with suspected infection. In the development of qSOFA, striking a balance between simplicity and accuracy was highlighted. To provide a parsimonious score for bedside use, qSOFA was originally developed with only three variables (i.e., systolic blood pressure, respiratory rate, and Glasgow Coma Scale) that are readily available in emergency departments; however, many retrospective studies suggested to augment qSOFA by adding a number of sepsis risk factors to improve its prognostic performance [5–12]. As qSOFA focuses on acute physiological values, the suggested risk scores include patient demographic variables (e.g., age [5] and comorbidities [1]), functional variables (e.g., frailty scores [6]), and laboratory results (e.g., lactate [1, 8–11] and procalcitonin levels [7, 12]).

This study aimed to assess the additional value of sepsis-related risk factors to improve the prognostic value of qSOFA. Prior literature [5–12] has limitations in the ability to address this issue comprehensively. First, to reflect the actual clinical settings, we should use the qSOFA measurement taken at the time the clinicians suspect sepsis in the emergency department, rather than using the worst qSOFA calculated in a certain time frame. Second, a dataset with various types of sepsis-related risk factors (e.g. demographic, functional, and laboratory) was necessary for comparison. Third, to account for the external validity, a multicenter study was deemed more appropriate.

In this study, we assessed the value of qSOFA (alone or combined with other sepsis-related mortality risk factors) in predicting in-hospital mortality. We performed a post-hoc analysis of the data of a multicenter prospective trial dataset targeting adult patients who arrived at the emergency department with a suspected infection.

## Materials and methods

### Design, setting, and participants

This study was a post-hoc analysis of data from the Japanese Association for Acute Medicine Sepsis Prognostication in Intensive Care Unit and Emergency Room (JAAM SPICE-ER) study (Trial registration: UMIN000027258) [7]. This multicenter, prospective cohort study included 34 emergency departments (December 2017 to February 2018); participating patients were registered in either university or non-university hospitals. Adult patients (≥16 years) were included in the study if they were suspected to have an infection during their stay in the emergency department, defined as need for administration of antibiotics, order of microbiological investigations, or imaging request to identify infection focus. Participants were to be hospitalized in one of the study hospitals or had died in the emergency department. Patients were excluded if they were not hospitalized or if they were transferred to a non-participating hospital.

### Ethics approval and consent to participate

The study protocol was reviewed and approved by the Research Ethics Committee of all participating institutions at the Japanese Association for Acute Medicine (JAAM) SPICE study group. Given the retrospective and anonymized nature of this study in routine care, the Ethics Committees waived the need for informed consent from the study participants. The Institutional Review Board of Hokkaido University, a leading institution in SPICE, approved this study (Approval No. 016–0385). All data were fully anonymized before any analysis. This post-hoc analysis was performed upon completion of the data collection in July 2018.

### Variables and outcomes

Data collection was performed as part of the routine clinical workup. Data were recorded by SPICE-ER site investigators, mostly senior staff emergency physicians, throughout patients' hospitalization. Missing or contradictory data, or outlier values were checked with each investigator on the SPICE committee request. Data of the following variables were collected: patient demographics, clinical characteristics, comorbidities, level of clinical frailty, and qSOFA score. Based on prior literature [5–12] and clinical availability at the emergency department, we chose the following sepsis-related mortality risk factors: age, Charlson comorbidity index (CCI), Clinical Frailty Scale (CFS) score, and lactate level [5–14]. As qSOFA focuses on acute physiological values, we chose those risk scores to represent different categories of sepsis mortality: age and Charlson comorbidity score for patient demographic variables; CFS for the functional variable, and lactate for the laboratory variable. To reflect real clinical settings, we collected the qSOFA score at the time when the infection was first suspected by the treating emergency physician. As such, we avoided using the most abnormal measurement in the emergency department. A Glasgow Coma Scale score of <15 was a qSOFA criterion for altered mental status. Study outcomes were in-hospital mortality during the emergency department visit or hospitalization [1, 2].

### Statistical analysis

Logistic regression was used to assess the predictive value of in-hospital mortality of the following six models: 1) qSOFA; 2) qSOFA+Age; 3) q SOFA+CFS; 4) qSOFA+CCI; 5) qSOFA+lactate levels; and 6) qSOFA+Age+CCI+CFS+lactate levels. To thoroughly compare each model, we used the fitted values in each logistic regression model to calculate the area under the receiver operating characteristic curve (AUC). As such, the predictive value of the qSOFA

score was assessed using its fitted value in the logistic regression model, rather than its predictive value using a cutoff of 2. With this approach, we avoided having to provide any arbitrary cutoff for other models. Clinical risk factors were used as continuous rather than categorical variables, as we suspected a linear correlation between each score and morality risk. We also calculated other key clinical statistics (i.e., sensitivity, specificity, positive predictive value, and negative predictive value) for each model at the cutoff of Youden's index, where the sum of sensitivity and specificity is the largest in the receiver operator characteristic curve. We used the Delong method to compare AUC values among the models. P-values of <0.05 were indicative of statistical significance. Following prior literature, we considered a test with an AUC greater than 0.9 as having high accuracy, while that of 0.7–0.9 indicated moderate accuracy, 0.5–0.7 indicated low accuracy, and 0.5 indicated a chance result. [13, 14] Given a low rate of missing data in the original cohort, no imputation procedure was performed in the primary analysis. Sensitivity analysis was performed with multiple imputations. A subgroup analysis was performed based on age to investigate the performance of each model in a relatively younger population (age < 70 years old). Descriptive statistics were reported as counts (proportions) for categorical variables and as mean (standard deviation, SD) for continuous variables. The requirement for informed consent was waived by the ethics committee due to the observational nature of this study. This study was conducted in accordance with the TRIPOD reporting guidelines [15]. All analyses were performed with R software, version 3.5.2 (The R Foundation for Statistical Computing, Vienna, Austria).

## Results

A total of 1060 patients with suspected infection were included during the study period. After excluding 109 patients with missing data (99 for lactate, 7 for qSOFA, and 3 for CFS), 951 were included in the analysis. In this cohort, the in-hospital mortality rate was 15.9%. Approximately 60% of the population were males in both the non-survivor and survivor groups (55.6% vs. 60%, p = 0.362). Non-survivors were older (mean age 80 vs. 75 years, p<0.001) and more frail (mean CFS 5 vs. 4, p<0.001) than survivors. Similarly, non-survivors had a higher mean CCI score (3 vs. 2 points, p = 0.141), higher mean qSOFA score (2 vs. 1 point, p<0.001), and higher lactate levels (4.7 vs. 2.5 mmoL/L, p<0.001) (Table 1) than survivors. Pulmonary sepsis was more frequent among non-survivors than among survivors (59.6% vs. 45.9%). The frequency of urinary sepsis was similar in both cohorts (12.6% vs. 13.8%). There was a linear correlation between in-hospital mortality rate and qSOFA score (Fig 1); a similar association was observed between in-hospital mortality rate and age, CFS score, and lactate levels. The correlation between the CCI score and mortality rate was weaker than that between mortality rate and other variables.

### Primary analysis

The AUC for the qSOFA-Only model predicting in-hospital mortality was 0.627 (95% confidence interval [CI]: 0.580–0.673) (Table 2). Addition of other variables marginally improved AUC values; the final model (qSOFA+Age+CCI+CFS+lactate level) yielded an AUC of 0.730 (95% CI: 0.687–0.774). Other key statistic values at Youden's index were similar among all models, with sensitivity and specificity in the range of 0.55–0.65 and 0.60–0.75, respectively. Using the Delong method, we compared the AUC between the qSOFA-Only model and other models. Both lactate levels and the CFS score significantly improved AUC values, but point estimate only marginally changed from 0.627 (qSOFA-Only) to 0.663 (qSOFA+CFS model) or to 0.695 (qSOFA+lactate model). Using our pre-defined criteria (AUC >0.9 was deemed to indicate high accuracy, while that of 0.7–0.9 indicated moderate accuracy, 0.5–0.7 indicated

**Table 1. Baseline characteristics of trauma patients stratified by survival outcomes during hospitalization.**

| Characteristics | Overall | Died | Survived | P-value |
|---|---|---|---|---|
| **N** | 951 | 151 | 800 | |
| **Sex (male, %)** | 564 (59.3) | 84 (55.6) | 480 (60.0) | 0.362 |
| **Age, years (median [IQR])** | 79 [68.5; 85] | 82 [74; 87] | 78 [68; 85] | <0.001 |
| **Clinical Frailty Scale (median [IQR])** | 4 [3; 6] | 5 [4; 7] | 4 [3; 6] | <0.001 |
| **Charlson comorbidity index (median [IQR])** | 1 [1; 2] | 2 [1; 4] | 2 [0; 3] | 0.062 |
| **qSOFA score (median [IQR])** | 1 [1; 2] | 2 [1; 2] | 1 [1; 2] | <0.001 |
| **Lactate, mmol/L (median [IQR])** | 1.9 [1.3; 3.2] | 3.2 [1.7, 6.2] | 1.8 [1.2; 2.8] | <0.001 |
| **Respiratory rate (median [IQR])** | 23 [18; 28] | 24 [19; 30] | 22 [18; 27] | 0.001 |
| **Systolic blood pressure, mmHg (mean [SD])** | 127 (33) | 115 (36) | 129 (31) | <0.001 |
| **Diastolic blood pressure, mmHg (mean [SD])** | 72 (20) | 67 (22) | 74 (20) | <0.001 |
| **Mean blood pressure, mmHg (mean [SD])** | 91 (23) | 83 (25) | 92 (22) | <0.001 |
| **Heart rate (mean [SD])** | 99 (25) | 99 (28) | 99 (24) | 0.738 |
| **GCS—Eyes (median [IQR])** | 4 [4; 6] | 4 [3; 4] | 4 [4; 4] | <0.001 |
| **GCS—Verbal (median [IQR])** | 4 [4; 5] | 4 [2; 5] | 5 [4; 5] | <0.001 |
| **GCS—Motor (median [IQR])** | 6 [6; 6] | 6 [4; 6] | 6 [6; 6] | <0.001 |
| **GCS—Total (median [IQR])** | 14[13; 15] | 13 [9; 15] | 14 [13; 15] | <0.001 |
| **Body temperature, Celsius (median [IQR])** | 37.5 [36.6; 38.5] | 37.00 [36.1; 38.0] | 37.6 [36.7, 38.6] | <0.001 |
| **Length of admission, days, (mean [SD])** | 23 (23) | 20 (29) | 23 (22) | 0.076 |
| **Site of infection (%)** | | | | <0.001 |
| **Respiratory** | 457 (48.1) | 90 (59.6) | 367 (45.9) | |
| **Abdominal** | 174 (18.3) | 14 (9.3) | 160 (20.0) | |
| **Urinary** | 129 (13.6) | 19 (12.6) | 110 (13.8) | |
| **Skin and soft tissue** | 43 (4.5) | 7 (4.6) | 36 (4.5) | |
| **Central nervous system** | 14 (1.5) | 0 (0.0) | 14 (1.8) | |
| **Bone and joint** | 7 (0.7) | 1 (0.7) | 6 (0.8) | |
| **Endocarditis** | 5 (0.5) | 4 (2.6) | 1 (0.1) | |
| **Catheter** | 3 (0.3) | 1 (0.7) | 2 (0.2) | |
| **Implant** | 2 (0.2) | 0 (0.0) | 2 (0.2) | |
| **Wounds** | 3 (0.3) | 1 (0.7) | 2 (0.2) | |
| **Others** | 51 (5.4) | 4 (2.6) | 47 (5.9) | |
| **Unknown** | 63 (6.6) | 10 (6.6) | 53 (6.6) | |
| **Discharge disposition (%)** | | | | <0.001 |
| **Died** | 151 (15.9) | 151 (100.0) | 0 (0.0) | |
| **Home** | 545 (57.3) | 0 (0.0) | 545 (68.1) | |
| **Institute** | 255 (26.8) | 0 (0.0) | 255 (31.9) | |

Abbreviation: GCS, Glasgow Coma Scale; SD, standard deviation; qSOFA, quick sepsis-related organ failure assessment

low accuracy, and 0.5 indicated a chance result [13, 14]), only the final model had an AUC slightly higher than the threshold of moderate accuracy, while all the other models were regarded as having low accuracy.

## Sensitivity analysis

In subgroup analysis of the younger cohort (Table 3), following sensitivity analysis with multiple imputation, the predictive performance of qSOFA was poor, regardless of the variables included in the combined model (S1 Table).

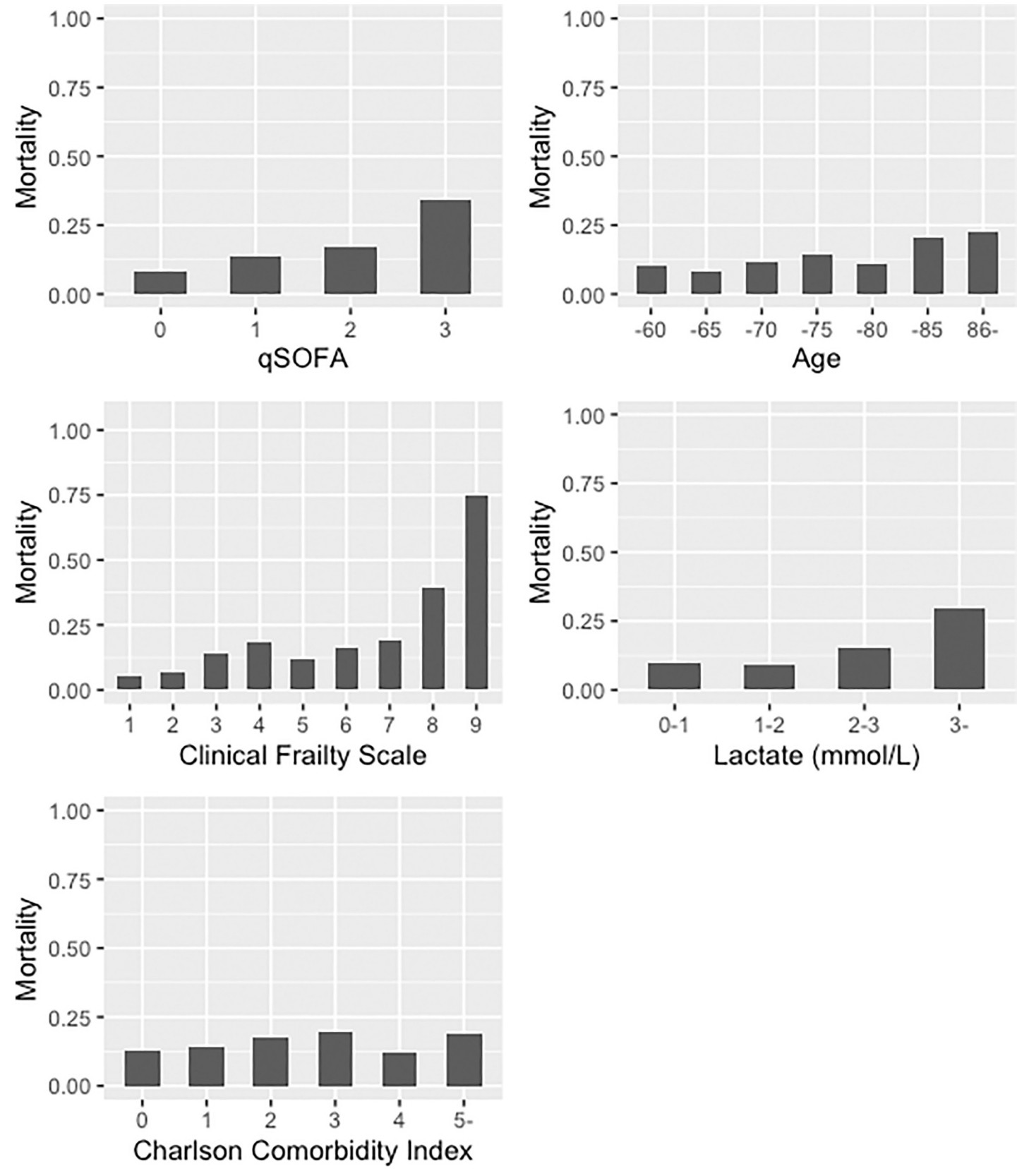

**Fig 1. Incidence of in-hospital mortality, stratified by clinical score types.**

**Table 2. Comparison of areas under the curve and other key clinical statistics between prediction models.**

| Model | AUC (95% CI) | p-value* |
|---|---|---|
| **qSOFA-Only** | 0.627 [0.580; 0.673] | reference |
| **qSOFA+Age** | 0.650 [0.602; 0.698] | 0.058 |
| **qSOFA+CCI** | 0.639 [0.592; 0.686] | 0.109 |
| **qSOFA+CFS** | 0.663 [0.617, 0.710] | 0.017 |
| **qSOFA+lactate** | 0.695 [0.646; 0.743] | <0.001 |
| **qSOFA+Age+CCI +CFS+lactate** | 0.730 [0.687; 0.774] | <0.001 |

| Model | Sensitivity (95% CI) | Specificity (95% CI) | Positive predictive value(95% CI) | Negative predictive value(95% CI) | Positive likelihood ratio(95% CI) | Negative likelihood ratio(95% CI) |
|---|---|---|---|---|---|---|
| **qSOFA only** | 0.55 [0.47; 0.63] | 0.63 [0.59; 0.66] | 0.22 [0.18; 0.26] | 0.88 [0.85; 0.91] | 1.48 [1.24; 1.75] | 0.72 [0.60; 0.86] |
| **qSOFA+Age** | 0.62 [0.54; 0.70] | 0.60 [0.57; 0.64] | 0.23 [0.19; 0.27] | 0.89 [0.87; 0.92] | 1.57 [1.35; 1.82] | 0.63 [0.51; 0.77] |
| **qSOFA+CCI** | 0.67 [0.59; 0.74] | 0.55 [0.51; 0.58] | 0.22 [0.18; 0.26] | 0.90 [0.87; 0.92] | 1.47[1.28; 1.68] | 0.61 [0.48; 0.77] |
| **qSOFA+CFS** | 0.60 [0.51; 0.68] | 0.66 [0.62; 0.69] | 0.25 [0.20; 0.29] | 0.90 [0.87; 0.92] | 1.73[1.47; 2.03] | 0.62 [0.50; 0.75] |
| **qSOFA+lactate** | 0.59 [0.51; 0.67] | 0.75 [0.72; 0.78] | 0.31 [0.25; 0.36] | 0.91 [0.88; 0.93] | 2.35[1.96; 2.81] | 0.55 [0.45; 0.67] |
| **qSOFA+Age+CCI +CFS+lactate** | 0.64 [0.56; 0.72] | 0.70 [0.67; 0.73] | 0.29 [0.24; 0.34] | 0.91 [0.89; 0.93] | 2.14[1.83; 2.51] | 0.51 [0.41; 0.64] |

*p-value: comparison with the AUC of qSOFA-Only (model 1)

Key clinical statistics calculated with the Youden's index, whereby the sum of sensitivity and specificity is the biggest. Abbreviations: AUC, area under the curve; CCI, Charlson comorbidity index; CFS, Clinical Frailty Scale; CI, confidence interval; NPV, negative predictive value; PPV, positive predictive value; qSOFA; quick sepsis-related organ failure assessment

## Discussion

In this retrospective analysis of a prospectively collected dataset, we investigated the prognostic accuracy of the qSOFA score, alone and combined with other risk factors, for mortality among patients presenting to the emergency department with a suspected infection. Addition of other variables only marginally improved AUC values; even the model with the best performance (qSOFA+Age+CCI+CFS+lactate levels) yielded an AUC of 0.730 (95% CI: 0.687−0.774) compared to that of 0.627 (95% CI: 0.580−0.673) observed with the qSOFA-Only model.

### Results of previous studies and interpretation of the present study findings

Several previous studies have assessed the use of the qSOFA score in predicting sepsis-related mortality in emergency care settings. Freund et al. conducted one of the largest studies that assessed the clinical utility of qSOFA in the emergency department [16]; they conducted a multicenter prospective cohort study with 30 emergency departments in Europe and recruited 879 patients in the analysis. In contrast to our study, the AUC of the qSOFA score for in-hospital mortality was 0.80 in their analysis. This discrepancy might be accounted for by methodological differences between that study and the present study. First, the median age of our cohort was 79 years, compared to that of 67 years in the European cohort. Vital signs vary

**Table 3. Comparison of areas under the curve and other key clinical statistics between prediction models for patients aged <70 years.**

| Models | AUC [95% CI] | p-value* | | | |
|---|---|---|---|---|---|
| **qSOFA-Only** | 0.579 [0.483; 0.675] | reference | | | |
| **qSOFA+Age** | 0.595 [0.495; 0.695] | 0.597 | | | |
| **qSOFA+CCI** | 0.591 [0.494; 0.689] | 0.600 | | | |
| **qSOFA+CFS** | 0.624 [0.523; 0.724] | 0.319 | | | |
| **qSOFA+lactate** | 0.718 [0.620; 0.816] | 0.025 | | | |
| **qSOFA+Age+CCI +CFS+lactate** | 0.724 [0.634; 0.813] | 0.015 | | | |
| **Model** | **Sensitivity(95% CI)** | **Specificity(95% CI)** | **Positive predictive value(95% CI)** | **Negative predictive value(95% CI)** | **Positive likelihood ratio(95% CI)** | **Negative likelihood ratio(95% CI)** |
| **qSOFA only** | 0.59 [0.49; 0.67] | 0.61 [0.56; 0.65] | 0.25 [0.20; 0.30] | 0.87 [0.83; 0.90] | 1.49[1.24; 1.78] | 0.68[0.55; 0.85] |
| **qSOFA+Age** | 0.50 [0.40; 0.59] | 0.73 [0.69; 0.77] | 0.29 [0.23; 0.36] | 0.87 [0.83; 0.90] | 1.83[1.46; 2.29] | 0.69[0.58; 0.83] |
| **qSOFA+CCI** | 0.65 [0.56; 0.73] | 0.58 [0.54; 0.62] | 0.26 [0.21; 0.31] | 0.88 [0.84; 0.91] | 1.55[1.32; 1.83] | 0.60[0.47; 0.77] |
| **qSOFA+CFS** | 0.66 [0.57; 0.74] | 0.60 [0.57; 0.65] | 0.27 [0.22; 0.32] | 0.89 [0.85; 0.92] | 1.67[1.41; 1.96] | 0.56[0.44; 0.73] |
| **qSOFA+lactate** | 0.60 [0.51; 0.69] | 0.75 [0.71; 0.79] | 0.35 [0.29; 0.42] | 0.89 [0.86; 0.92] | 2.43[1.98; 2.98] | 0.53[0.42; 0.66] |
| **qSOFA+Age+CCI +CFS+lactate** | 0.66 [0.57; 0.74] | 0.68 [0.64; 0.72] | 0.32 [0.26; 0.38] | 0.90 [0.87; 0.93] | 2.07[1.74; 2.47] | 0.50[0.39; 0.64] |

*p-value: comparison with the AUC of qSOFA-only (Model 1)

Key clinical statistics calculated with the Youden's index, whereby the sum of sensitivity and specificity is the biggest. Abbreviations: AUC, area under the curve; CCI, Charlson comorbidity index; CFS, Clinical Frailty Scale; CI, confidence interval; NPV, negative predictive value; PPV, positive predictive value; qSOFA; quick sepsis-related organ failure assessment

among age groups; as a result, the prognostic value of the qSOFA score might be higher in the younger population. However, in our subgroup analysis, the AUC was still not as high as 0.80 among patients aged <70 years. Second, our cohort had a urinary sepsis rate of only 15% as compared to a rate of 27% in the European study. Urinary sepsis is associated with better prognosis than that observed with other types of sepsis, which may explain the overall mortality of 15% in our cohort, compared to that of 7% in their cohort. Third, and most importantly, our study calculated the qSOFA score when a physician suspected an infection, rather than calculating the worst qSOFA score during the patient's stay in the emergency department. We believe that to represent a real-world setting, qSOFA values of interest should be those obtained at the time when the infection is suspected.

There are a few smaller studies that aimed to improve the predictive value of qSOFA in the ED setting. In a single center prospective observational study from Australia [17], qSOFA was combined with CCI to improve its performance to predict 28day mortality. Similar to our study, the AUC only marginally increased from 0.72−0.79. (95% CI 0.62−0.82 vs. 0.71−0.88, P = 0.055). Another study from Indonesia [18] added lactate to qSOFA, and assessed its performance to predict the in-hospital mortality. The AUC was slightly better than the qSOFA only model (0.74 vs. 0.70, 95% CI 0.71−0.77 vs. 0.67−0.74; p = 0.006). A retrospective cohort study from Korea [19] investigated the additional value of the red cell distribution width to qSOFA to predict the 30-day mortality, which has also marginally improved the AUC (0.71 vs. 0.66, 95% CI 0.69−0,74 vs. 0.63−0.68). All of the above studies aimed to improve the qSOFA by adding readily available values in the ED, all of which resulted in a subtle improvement in the predictive value of qSOFA. The present study findings are consistent with those of previous

studies, demonstrating poor prognostic value of the qSOFA score for sepsis-related mortality despite model augmentation with other risk factors.

Although, our results were not in favor of complicating qSOFA with any additional values, a prospective observational trial from the United States [20] successfully improved the qSOFA by adding the clinician's "gut feeling" for the 28-day mortality. Despite being subjective, such a prediction based on the ED physician's experience may be one of the most important surrogate markers for the patients' risk of death. In the ED, we always process and synergize information (e.g. patient background, past histories, social status, exam findings, laboratory results, and patient wishes) to make a clinical prediction. Therefore, it is possible that the addition of any such variable may somehow improve a simple score like qSOFA, as proven in this trial. However, we used all the above mentioned information to predict mortality for patients in the emergency department in our study. Over-simplification of such complex and multifactorial processes by making a simple prediction rule on qSOFA may be extremely difficult. Therefore, clinicians should be cautious in solely relying on the qSOFA score for such predictions.

## Study strengths

This study has several strengths. First, we used a dataset prospectively collected at 32 hospitals, including both university and non-university hospitals; as a result, the presented findings have high external validity for Japan.

Second, we calculated the qSOFA score at the time when the attending physician suspected the infection rather than using the worst qSOFA score recorded during the patient's stay in the emergency department; this approach is unique to our study, as it requires prospective data collection. In addition, use of the values obtained at the time of infection suspicion reflects the real-world decision-making process. Third, we assessed the prognostic value of the qSOFA score in the emergency department rather than in the ICU setting. As the qSOFA score was developed for use in the emergency department setting, its validation should be performed on data from patients treated in this setting. Fourth, we included patients based on the attending physician's suspicion of infection rather than based on retrospective medical chart review or presence of specific diagnostic codes, which is better than other methods at reflecting the real-world clinical setting.

## Study limitations

First, this study was based in Japan, which is a developed country with an aging society; however, external validity of the present findings and their generalizability to other populations remain unclear.

Second, the accuracy of the dataset is unclear. For example, we were unable to provide any training programs for each investigator to calculate the CFS. Such training might have improved data accuracy; nevertheless, distribution of variables of interest did not raise bias concerns (Table 1 and Fig 1).

Third, our study did not account for treatment differences among patients, which might have affected their in-hospital mortality risk. However, this study aimed to assess a predictive model of in-hospital mortality based on variables collected during emergency department admission.

Fourth, we did not exclude patients with treatment limitations, which could have skewed the reported mortality rate.

Fifth, we compared the sensitivity and specificity of each score at Youden's index, where the sum of sensitivity and specificity is at its max. We were unable to calculate and compare the sensitivity and specificity at a given threshold, such as 2 for qSOFA. This enabled us to

compare the performance of each model without using any arbitrary threshold for each model (e.g., the best threshold for the qSOFA+age model is unknown). Rather than providing such arbitrary thresholds, we aimed to use a common threshold and compared key clinical statistics.

Finally, this was a post-hoc analysis of a prospectively collected dataset. As a result, we were unable to publish the study protocol ahead of data collection. Finally, the lack of long-term mortality data was a limitation of our dataset.

## Future implications

Given that qSOFA was not originally developed as a highly accurate prognostic tool, the use of qSOFA may not be suitable for the prediction of mortality regardless of the augmentation. The present study reinforced this hypothesis using a multicenter prospective trial dataset. Mortality prediction for a patient in the emergency department is multifactorial and complex in nature; therefore, clinicians should be cautious in solely relying on the qSOFA score.

## Conclusions

In this post-hoc analysis of data from a prospective multicenter study in Japan, qSOFA failed to adequately predict sepsis-related in-hospital mortality among patients presenting at an emergency department. Combining the qSOFA score with other risk factors for mortality, including age, CFS score, CCI, and lactate levels, only marginally improved the prognostic value of the model.

## Supporting information

**S1 Table. Comparison of AUCs in each prediction model with multiple imputation.**
(DOCX)

**S2 Table. Comparison of OR and VIF amongst each model.**
(DOCX)

**S3 Table. Subgroup analysis based on the microbiological evidence of the infection.**
(DOCX)

## Acknowledgments

We thank the JAAM SPICE Study Group for their contribution to this study.

## Author Contributions

**Supervision:** Atsushi Shiraishi, Satoshi Gando, Toshikazu Abe, Shigeki Kushimoto, Toshihiko Mayumi, Seitaro Fujishima, Akiyoshi Hagiwara, Toru Hifumi, Akira Endo, Takayuki Komatsu, Joji Kotani, Kohji Okamoto, Junichi Sasaki, Yasukazu Shiino, Yutaka Umemura.

**Writing – original draft:** Ryo Ueno.

**Writing – review & editing:** Takateru Masubuchi.

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
