## [Decision Letter · Decision Letter 0]

22 Mar 2021

PONE-D-20-37481

Quick sequential organ failure assessment score combined with other sepsis-related risk factors to predict in-hospital mortality: Post-hoc analysis of prospective multicenter study data

PLOS ONE

Dear Dr. Shiraishi,

Thank you for submitting your manuscript to PLOS ONE. After careful consideration, we feel that it has merit but does not fully meet PLOS ONE’s publication criteria as it currently stands. Therefore, we invite you to submit a revised version of the manuscript that addresses the points raised during the review process.

We look forward to receiving your revised manuscript.

Kind regards,

Juan Carlos Lopez-Delgado, MD, PhD

Academic Editor

PLOS ONE

Additional Editor Comments:

Dear authors,

All apologies for the late reply. It was very difficult to find appropiate reviewers for such important work during COVID-19 pandemia.

I expect our decision will not discourage to resubmit an improved version of your manuscript.

Warm regards & Keep safe!

Juan Carlos

Journal Requirements:

2. In the ethics statement in the manuscript and in the online submission form, please provide additional information about the patient records/samples used in your retrospective study, including: a) whether all data were fully anonymized before you accessed them; b) the date range (month and year) during which patients' medical records/samples were accessed c) the source of the medical records/samples analyzed in this work (e.g. hospital, institution or medical center name).

"This study did not receive any specific grant from funding agencies in the public, commercial, or not-for-profit

sectors. RU is supported by the Masason Foundation (MF) and has received a grant from MF. The funding body

played no role in the study design; data collection, management, analysis, or interpretation; manuscript

preparation; or the decision to submit the report for publication."

"The authors received no specific funding for this work."

Reviewers' comments:

Reviewer's Responses to Questions

**Comments to the Author**

1. Is the manuscript technically sound, and do the data support the conclusions?

Reviewer #1: Yes

Reviewer #2: Partly

2. Has the statistical analysis been performed appropriately and rigorously? 

Reviewer #1: No

Reviewer #2: Yes

3. Have the authors made all data underlying the findings in their manuscript fully available?

Reviewer #1: No

Reviewer #2: Yes

4. Is the manuscript presented in an intelligible fashion and written in standard English?

Reviewer #1: Yes

Reviewer #2: Yes

5. Review Comments to the Author

Reviewer #1: The study analyzes a group of patients admitted to 34 Japanese hospitals for 2 years on suspicion of infection. The value of the qSOFA scale is calculated at the time of suspected infection and the qSOFA calculation is added along with other models: 1) qSOFA; 2) qSOFA+Age; 3) q SOFA+CFS; 4) qSOFA+CCI; 5) qSOFA+lactate levels; 6) qSOFA+Age+CCI+CFS+lactate levels. The study is very interesting, but I think it presents several problems. On the one hand, they do not calculate the sample size, I consider that they do not calculate the sample size. They would have to explain how they calculate the analyzed models. On the other hand, they would have to improve the statistical analyses they perform.

Introduction:

I think the introduction is correct.

Material and methods.

The authors would have to perform a sample size calculation. They would have to specify how they calculate each of the analyzed models. They would have to do a normality study of the quantitative variables analyzed. They would also have to calculate in addition to sensitivity, specificity, positive and negative predictive values likehood positive and negative ratios with their respective confidence intervals at 95%

Results.

The results are adequate but the results of the sensitivity, specificity, PPV and NPV that the authors give should give them with 95% confidence intervals. They would also have to calculate positive and negative likehood ratios. Both in the global analysis and in the analysis that make the youngest population below 70 years. I am drawn to the low number of patients recruited from 32 hospitals over two years during the study, the authors should have calculated a sample size.

Discussion.

The authors should improve the discussion by only writing about the usefulness of the qSOFA scale but do not affect the results obtained by the rest of the models studied. They would have to better explain the results obtained from each model and because the models improve, even if it is slightly to the qSOFA score.

The authors tell us that there are several studies that analyze the result of qSOFA but only analyze its results against 2 of them and especially one of them, I think the authors would have to expand the number of articles with which to compare their results.

Reviewer #2: This study aims to analyze a the value of the qSOFA scale in patients of suspicion of infection in a multicenter approach.

However, I have the feeling the data is underused from the statistical point of view and some modifications should be performed before publication.

Major concerns,

- Please, confirm in methods (or not) if all the patients suffer from infection. qSOFA should be applied to those with confirmed infection since it was designed to predict infection.

- I recommend to analyze those if there are differences in prediction of mortality in those with confirmed infection and those without.

- I recommed to analyze risk factors for mortality to elucidate if qSOFA represent a variable associated with mortality.

Minor concerns:

- Disccuss more in deep your results with those studies from the literature. Expand references.

6. PLOS authors have the option to publish the peer review history of their article (what does this mean?). If published, this will include your full peer review and any attached files.

Reviewer #1: No

Reviewer #2: No

---

## [Author Response · Author response to Decision Letter 0]

27 May 2021

We have attached a rebuttal letter for your perusal. We sincerely thank the reviewers and the editor for their time and effort.

---

## [Decision Letter · Decision Letter 1]

25 Jun 2021

Quick sequential organ failure assessment score combined with other sepsis-related risk factors to predict in-hospital mortality: Post-hoc analysis of prospective multicenter study data

PONE-D-20-37481R1

Dear Dr. Shiraishi,

We’re pleased to inform you that your manuscript has been judged scientifically suitable for publication and will be formally accepted for publication once it meets all outstanding technical requirements.

Kind regards,

Juan Carlos Lopez-Delgado, MD, PhD

Academic Editor

PLOS ONE

Additional Editor Comments (optional):

Dear authors,

Please, keep in mind the written english should be correct. This is important since any of the reviewers is a native english speaker and PLOS One does not provide any review before publication.

Regards,

Juan Carlos

Reviewers' comments:

Reviewer's Responses to Questions

**Comments to the Author**

1. If the authors have adequately addressed your comments raised in a previous round of review and you feel that this manuscript is now acceptable for publication, you may indicate that here to bypass the “Comments to the Author” section, enter your conflict of interest statement in the “Confidential to Editor” section, and submit your "Accept" recommendation.

Reviewer #1: All comments have been addressed

Reviewer #2: All comments have been addressed

2. Is the manuscript technically sound, and do the data support the conclusions?

Reviewer #1: Yes

Reviewer #2: Yes

3. Has the statistical analysis been performed appropriately and rigorously? 

Reviewer #1: Yes

Reviewer #2: Yes

4. Have the authors made all data underlying the findings in their manuscript fully available?

Reviewer #1: Yes

Reviewer #2: Yes

5. Is the manuscript presented in an intelligible fashion and written in standard English?

Reviewer #1: Yes

Reviewer #2: Yes

6. Review Comments to the Author

Reviewer #1: I believe that the article has improved, the authors have answered my questions and suggestions in a satisfactory way.

Reviewer #2: Authors have addressed all the pending questions. Manuscript has improved a lot in comparison with the first version and it can be already published.

7. PLOS authors have the option to publish the peer review history of their article (what does this mean?). If published, this will include your full peer review and any attached files.

Reviewer #1: No

Reviewer #2: No

---

## [Editor Report · Acceptance letter]

6 Jul 2021

PONE-D-20-37481R1 

Quick sequential organ failure assessment score combined with other sepsis-related risk factors to predict in-hospital mortality: Post-hoc analysis of prospective multicenter study data 

Dear Dr. Shiraishi:

I'm pleased to inform you that your manuscript has been deemed suitable for publication in PLOS ONE. Congratulations! Your manuscript is now with our production department. 

Kind regards, 

on behalf of

Dr. Juan Carlos Lopez-Delgado 

Academic Editor

PLOS ONE